# Transcriptome Analysis Revealed That Hydrogen Peroxide-Regulated Oxidative Phosphorylation Plays an Important Role in the Formation of *Pleurotus ostreatus* Cap Color

**DOI:** 10.3390/jof9080823

**Published:** 2023-08-03

**Authors:** Ludan Hou, Kexing Yan, Shuai Dong, Lifeng Guo, Jingyu Liu, Shurong Wang, Mingchang Chang, Junlong Meng

**Affiliations:** 1College of Food Science and Engineering, Shanxi Agricultural University, Taigu 030801, China; houludan@126.com (L.H.); yan15934475351@163.com (K.Y.); m15513733589@163.com (S.D.); m17836509709@163.com (L.G.); liujingyu80@126.com (J.L.); wzlj2005@163.com (S.W.); 2Shanxi Key Laboratory of Edible Fungi for Loess Plateau, Taigu 030801, China; 3Shanxi Research Center for Engineering Technology of Edible Fungi, Taigu 030801, China

**Keywords:** *Pleurotus ostreatus*, hydrogen peroxide, cap color, oxidative phosphorylation, respiratory chain

## Abstract

*Pleurotus ostreatus* is widely cultivated in China. H_2_O_2_, as a signaling molecule, can regulate the formation of cap color, but its regulatory pathway is still unclear, severely inhibiting the breeding of dark-colored strains. In this study, 614 DEGs specifically regulated by H_2_O_2_ were identified by RNA-seq analysis. GO-enrichment analysis shows that DEGs can be significantly enriched in multiple pathways related to ATP synthesis, mainly including proton-transporting ATP synthesis complex, coupling factor F(o), ATP biosynthetic process, nucleoside triphosphate metabolic processes, ATP metabolic process, purine nucleoside triphosphate biosynthetic and metabolic processes, and purine ribonuclease triphosphate biosynthetic metabolic processes. Further KEGG analysis revealed that 23 DEGs were involved in cap color formation through the oxidative phosphorylation pathway. They were enriched in Complexes I, III, IV, and V in the respiratory chain. Further addition of exogenous uncoupling agents and ATP synthase inhibitors clarifies the important role of ATP synthesis in color formation. In summary, H_2_O_2_ may upregulate the expression of complex-encoding genes in the respiratory chain and promote ATP synthesis, thereby affecting the formation of cap color. The results of this study lay the foundation for the breeding of dark-colored strains of *P. ostreatus* and provide a basis for the color-formation mechanism of edible fungi.

## 1. Introduction

Fungal melanin is a high-molecular weight pigment with multiple biological functions. At present, with increasing attention being paid to food safety, natural pigments have received widespread attention. Studies have shown that fungal melanin is a potential antibacterial agent [1]. For example, *Schizophyllum commune* melanin has significant antibacterial activity against multidrug-resistant pathogens [2]. *Streptomyces glaucescens* melanin has in vitro anticancer activity against skin cancer cells. Furthermore, melanin is a highly effective radiation protective agent, and its radiation protection mechanism may be related to regulating survival-promoting signals, preventing oxidative stress, and regulate immunity. In addition, natural melanin also has anti-inflammatory activity and hypoglycemic and antihyperlipidemic effects [3]. It also has potential as a functional food [4]. Meanwhile, melanin may be useful as a potent agent for the prevention and management of high-fat diet-induced hyperlipidemia [5] with significant immunoregulatory activities, and it might be a promising source of immunoregulators in the healthcare field [6].

Recently, as an important source of natural pigments, research on the color-related properties of edible mushrooms has gradually begun. At present, various edible mushroom pigments have been extracted and identified. For example, carotenoids from *Cordyceps militaris* [7], red pigments from *Lactarius lilacinus* [8], and melanins from *Auricularia heimuer* [9] have all been studied and reported. Due to people’s desire for health, edible mushrooms are popular due to their functional biomolecules and biological activities. Therefore, research on the function of edible mushroom melanin has also been gradually carried out. For example, *A. heimuer* melanin has a significant effect on the treatment of alcoholic liver injury in vitro and in vivo, which may be an effective strategy to alleviate alcohol-induced liver injury [10]. The application of omics technology in the field of edible mushroom research has promoted in-depth research on the melanin synthesis pathway of edible mushrooms. For example, 26 genes were selected from the *Agaricus bisporus* H97 genome to participate in melanin synthesis through omics analysis [11]. Cytochrome p450 plays an important role in the formation of the cap color of *Hypsizygus marmoreus*. The L-DOPA pathway is the main pathway of melanin synthesis in the cap according to transcriptome analysis [12].

*P. ostreatus,* as a typical heterothallism edible fungus, is widely cultivated all over the world [13,14] and has the potential to be a model organism of edible mushrooms. In addition, *P. ostreatus* is rich in nutrients and contains various bioactive substances. It has various functions, such as antitumor, antioxidant, anti-inflammatory, and antiviral activities [15]. With the development of the big health industry, “black food” is favored by consumers because it contains mang anthocyanidins, proteins, vitamins, and other substances [16]. The color depth of the mushroom cap has become an important factor affecting consumers’ choices. From the perspective of the consumer market, the breeding of dark-colored varieties is an inevitable choice to improve industrial efficiency [17]. Cap color is an important agronomic trait that varies greatly among different varieties. Second, the color difference of the fruiting body cap will also be affected by various environmental factors. Current research shows that melanin mainly exists in three different forms: true melanin, brown melanin, and allomelanin [18,19]. In *Pleurotus* spp., the change in the color of the fruiting body cap is caused by the different relative ratios of true melanin and brown melanin [20]. However, the molecular mechanism is still unknown, which seriously restricts the breeding of dark-colored varieties of edible mushrooms.

H_2_O_2_, as a signaling molecule, can participate in various intracellular metabolic processes. Research has shown that H_2_O_2_ can participate in plant growth and morphogenesis, including seed germination, root gravitropism, and secondary wall differentiation [21,22,23]. Moreover, H_2_O_2_ is an important signaling molecule that can mediate stomatal closure and gene expression in the abscisic acid response, as well as UV-B-induced isoflavone accumulation [24,25]. In addition, H_2_O_2_ also plays an important role in the synthesis of melanin. Previous studies have shown that when H_2_O_2_ is present, cytochrome c can oxidize catecholamines and their cysteine derivatives, ultimately producing melanin. In contrast, when H_2_O_2_ is lacking, melanin cannot be formed [26]. This indicates the importance of H_2_O_2_ in the melanin-synthesis pathway. In addition, studies have shown that H_2_O_2_ can activate melanogenesis-related proteins, including cAMP responsive element-binding proteins, microphthalmia-related transcription factors, tyrosinase, and phenylalanine hydroxylase, thereby promoting melanogenesis [27,28]. Furthermore, H_2_O_2_ can increase the expression of ATP5B, intracellular ATP, and cAMP levels, thereby increasing the expression of melanin-promoting proteins and cellular melanogenesis [29]. In fungi, H_2_O_2_ can also serve as a signaling molecule to regulate fungal growth and development [30]. However, whether H_2_O_2_ can affect fungal color formation has not been reported.

At present, the mature construction of genetic transformation systems in edible mushrooms promotes the study of gene function and regulatory mechanisms. For example, research has been conducted on the regulatory mechanism of ganoderic acid biosynthesis in *G. lucidum* [31], the response mechanism to heat stress in *Pleurotus ostreatus* [32,33,34], and the growth and development mechanism of *Flammulina filiformis* [35,36]. This establishes a foundation for in-depth exploration of the molecular mechanism of color formation in edible mushrooms. Our previous research found that H_2_O_2_ can affect the expression of key genes in the melanin synthesis pathway, thereby affecting the formation of cap color [37]. In this study, RNA-seq-based transcriptome analysis was applied to further investigate the molecular mechanism by which H_2_O_2_ affects the formation of cap color. Our results revealed a global transcriptional response and further illustrated that the oxidative phosphorylation pathway may play a key role in the formation of cap color.

## 2. Materials and Methods

### 2.1. Strain

The strain CCMSSC 00389 of *P. ostreatus* was provided by the China Center for Mushroom Spawn Standards and Control and stored in the germplasm resource bank of Shanxi Agricultural University. The genome of strain CCMSSC 00389 has been published and can be obtained from DDBJ/EMBL/GenBank under the registration number MAYC0000000 [38].

### 2.2. Experiment with the Addition of Exogenous H_2_O_2_ or N,N′-dimethylthiourea (DMTU)

In this study, mushroom culture materials were prepared, bottled (200 g per bottle), and sterilized at 126 °C for 2 h for mushroom production. According to previous research reports [33], cultivation bottles were grouped and treated with an exogenous addition of H_2_O_2_ and DMTU. The bottles of different groups were processed once a day, and the changes in cap color were observed.

### 2.3. RNA Extraction, cDNA Library Construction, and RNA-seq

To understand the molecular mechanism through which H_2_O_2_ acts as a signaling molecule affecting the formation of cap color, young fruiting body samples of different treatment groups (control, 25 mM H_2_O_2_, and 50 mM DMTU) were established for RNA-seq. In this study, the total RNA of young fruiting bodies was extracted using TRIzol^®^ Reagent (Invitrogen, Carlsbad, CA, USA), and contaminating DNA was removed using DNase I (TaKaRa, Kyoto, Japan). Then, 1 μg of total RNA samples was processed using the TruSeqTM RNA sample preparation kit from Illumina (San Diego, CA, USA) to construct the RNA-seq transcriptome library. The RNA-seq library was sequenced with the Illumina NovaSeq 6000 sequencer. The sequencing data were deposited into the Sequence Read Archive of the National Center for Biotechnology Information (NCBI) with the accession number SUB13511634.

### 2.4. Analysis of DEGs

The fragments per kilobase of exon per million mapped reads (FRKM) method was used to normalize the expression levels of genes between the control, H_2_O_2_, and DMTU groups. The gene abundances were quantified using the expression quantification software RSEM. Then, DEGSeq2 was used to compare the different gene expression levels between different sample groups and screen for DEGs.

### 2.5. Bioinformatics Analysis

To further explore the regulatory pathways influenced by H_2_O_2_, Gene Ontology (GO) enrichment analysis was conducted on DEGs using the software Goatools to obtain GO functions that can be enriched by DEGs. Second, enrichment analysis of DEGs was conducted through the Kyoto Encyclopedia of Genes and Genomes (KEGG) database (http://www.genome.jp/kegg/, accessed on 28 December 2022) to obtain the regulatory pathways in which DEGs were enriched. Then, the connection between DEGs was analyzed based on the correlation between gene expressions.

### 2.6. Quantitative Real-Time PCR (qPCR)

Total RNA was extracted from different samples using the E.Z.N.A. plant RNA Kit (Omega Bio-Tek, Norcross, GA, USA) and subsequently converted to cDNA. In addition, the expression of DEGs in different samples was detected through qPCR by a ChamQTM SYBR qPCR Master Mix Kit (Vazyme, Nanjing, China). In this study, the *β*-*actin* gene and *β*-*tubulin* gene were used as references, and the relative expression of the gene was calculated according to the 2^−ΔΔCT^ method [34]. The primers used in this study for qPCR are shown in Appendix A.

### 2.7. Experiment with the Addition of Exogenous Oligomycin A or Valinomycin

Based on RNA-seq analysis, we further explored the factors that affect the formation of fungal cap color by spraying exogenous ATP synthase inhibitors (Oligomycin A) and uncoupling agents (valinomycin). According to the previous method, the cultivation bottles were prepared and incubated and then cultured at 25 °C in the dark. After the mycelium grows fully in the culture bottle, the bottles were transferred to the mushroom production box for production experiments. After the formation of young fruiting bodies, the bottles were divided into five groups with 10 bottles in each group. Considering that both Oligomycin A and valinomycin are insoluble in water, 10% ethanol was used for dissolution in this experiment. Therefore, 10% ethanol was established as the control group, and the other two groups were sprayed with 25 mM or 50 mM valinomycin externally. The last two groups were sprayed with 25 mM or 50 mM Oligomycin A externally.

### 2.8. Data Analysis

The experimental data were analyzed by one-way ANOVA according to Duncan’s test by SPSS software. GraphPad Prism 6 and photoshop software were used for figure analysis.

## 3. Results

### 3.1. H_2_O_2_ Regulates Cap Color Formation in P. ostreatus

To determine whether H_2_O_2_ regulates cap color formation in *P. ostreatus*, we treated the *P. ostreatus* cap with DMTU, which is a highly permeable molecule and an H_2_O_2_ scavenger that can reduce body damage in various biological systems. Figure 1 shows that exogenous spraying of H_2_O_2_ and DMTU after the formation of the primordia can significantly affect the formation of the cap color of *P. ostreatus*. In the mushroom production experiment, the results at 27 d and 29 d showed a visible darkening of the cap color after the external application of H_2_O_2_. In contrast, the addition of exogenous DMTU causes the cap color to turn white. This indicates that H_2_O_2_ may play a crucial role in the formation of the cap color of *P. ostreatus*.

### 3.2. The Oxidative Phosphorylation Pathway Plays a Key Role in Cap Color Formation

To identify the global transcriptomic changes in response to H_2_O_2_ or DMTU treatment, RNA-seq analysis was conducted. Treatment with exogenous DMTU hindered the formation of the cap color. The results showed that after the external application of DMTU, 1242 downregulated differentially expressed genes (DEGs) and 179 upregulated DEGs were identified from the control versus DMTU comparison (Figure 2). Figure 2 shows that there were 986 downregulated DEGs and 405 upregulated DEGs in the H_2_O_2_ experimental group compared with the DMTU experimental group. Among them, 314 upregulated DEGs and 300 downregulated DEGs were specifically regulated by H_2_O_2_ and DMTU. Therefore, the functions of 614 DEGs identified from the H_2_O_2_ versus DMTU comparison were examined to elucidate the possible mechanism through which H_2_O_2_-signaling molecules regulate the formation of the cap color.

To understand the functions of the DEGs identified from the H_2_O_2_ versus DMTU comparison, GO pathway enrichment analysis was performed (Figure 3A). The results showed that DEGs were significantly enriched in 18 pathways, mainly concentrated in the following pathways: proton-transporting ATP synthase complex, coupling factor F(o), ATP biosynthetic process, nucleoside triphosphate metabolic processes, ATP metabolic process, purine nucleoside triphosphate biosynthetic and metabolic processes, and purine ribonucleoside triphosphate biosynthetic metabolic processes. These pathways are closely related to ATP biosynthesis, so it is speculated that ATP synthesis and metabolism may be related to the formation of cap color. In addition, some important pathways were also significantly enriched in the following: inorganic ion transmembrane transport, ion transport, cation transmembrane transport, inorganic cation transmembrane transport, and proton transmembrane transporter activity. These pathways are closely related to the transport of ions in the mitochondrial respiratory chain. Furthermore, DEGs were significantly enriched in pathways such as the inner mitochondrial membrane protein complex and mitochondrial protein-containing complex (Figure 3A). Figure 3B shows that 20 key DEGs were further screened through a GO Enrichment Chord Chart. The results showed that 20 key genes can be found in multiple pathways, such as aerobic respiration, cellular respiration, aerobic electron transport chain, aerobic respiratory electron transport chain, and electron transport chain.

In the KEGG annotation analysis, the DEGs identified from the H_2_O_2_ versus DMTU comparison were classified into five categories: “Environmental Information Processing”, “organismal systems”, “molecular function”, “cellular processes”, “genetic information processing”, and “metabolism”. Within the five categories, 28 DEGs were collected and enriched in the energy metabolism pathway (Figure 4A). To further determine the functions of DEGs, a KEGG function enrichment analysis was performed. The results showed that the DEGs were significantly enriched in one metabolic pathway: oxidative phosphorylation. In addition, 23 DEGs were enriched in this metabolic pathway. More interestingly, after comparison, it is found that the differential genes in energy metabolism in Figure 4A and oxidative phosphorylation in Figure 4B are the same. Twenty-three DEGs are shown in Table 1.

In conclusion, oxidative phosphorylation and energy metabolism may be very important metabolic pathways in the formation of *P. ostreatus* cap color and can be regulated by the signaling molecule H_2_O_2_.

### 3.3. H_2_O_2_ Regulates Cap Color Formation by Affecting the Respiratory Chain

To further analyze the molecular mechanism by which H_2_O_2_ regulates the formation of the cap color, the functions of 23 DEGs were further analyzed. The results are shown in Table 1. The results showed that all 23 DEGs encoded subunits of complexes in the respiratory chain. In addition, the expression pattern of the 23 DEGs after the addition of H_2_O_2_ or DMTU was further analyzed with a heatmap. As shown in Figure 5, 23 DEGs were significantly upregulated under the action of exogenous H_2_O_2_. In contrast, DEGs were significantly downregulated after DMTU treatment. This further indicates that H_2_O_2_ may affect the formation of the cap color by upregulating the expression of key genes in the respiratory chain.

Furthermore, the pathway ID is map00190. As shown in Figure 6, 23 differentially expressed genes were enriched in Complexes I, III, IV, and V in the respiratory chain. Figure 7 shows that two DEGs (*g400*, *g8839*) participate in the encoding of Complex I; three DEGs participate in the encoding of Complex III; six DEGs participate in the encoding of Complex IV; and 12 DEGs participate in the encoding of Complex V. Figure 7 shows that there was a close connection between the 23 DEGs, with a high correlation between each gene.

To further verify the reliability of the transcriptome data, the relative gene expression of DEGs in each complex of the respiratory chain was detected. As shown in Figure 8A, in Complex I, compared with the control group, the relative expression levels of *g8849* and *g400* increased by 1.30- and 1.34-fold, respectively, in the exogenous H_2_O_2_ group and decreased by 40.9% and 62.81% in the DMTU group. In Complex III, compared to the control, the relative expression levels of *g2860*, *g13251*, and *g2546* in the H_2_O_2_ group were significantly increased by 1.57-, 1.40-, and 1.19-fold, respectively. In the DMTU group, the expression levels of *g2860* and *g13251* were downregulated by 43.54% and 39.31%, respectively, with *g2546* showing no significant changes compared to the control (Figure 8B). Figure 8C shows that six DEGs were involved in the encoding of Complex IV. Compared with the control group, the expression levels of all six DEGs in the H_2_O_2_ group were significantly upregulated. In the H_2_O_2_ scavenging experimental group (DMTU), all six DEGs were significantly downregulated. This indicated that the H_2_O_2_-signaling molecule can regulate the expression of the encoding genes of Complex IV subunits. Figure 8D shows that 12 DEGs were involved in the encoding of Complex V (ATP synthase). Under the regulation of the signal molecule H_2_O_2_, only the relative expression level of *g12225* showed no significant difference compared to the control, and the expression levels of 11 DEGs were significantly upregulated. However, after treatment with exogenous DMTU (H_2_O_2_ scavenger), the relative expression levels of 12 DEGs were significantly inhibited. The above results indicated that H_2_O_2_ plays an important regulatory role in the encoding of ATP synthase.

In summary, the signaling molecule H_2_O_2_ may participate in the formation of cap color by upregulating the expression of genes encoding Complexes I, III, IV, and V in the respiratory chain.

### 3.4. ATP Synthesis Affects the Formation of P. ostreatus Cap Color

Subsequently, valinomycin is a type of uncoupling agent that can dissipate the energy generated by electron transfer in the respiratory chain in the form of heat. To explore the factors that affect color formation by spraying exogenous valinomycin. The results showed that the addition of exogenous valinomycin resulted in a lighter cap color compared to the control group, indicating that ATP synthesis was hindered and affected by the formation of cap color (Figure 9). Furthermore, we determined the important role of ATP in the formation of cap color by adding exogenous Oligomycin A (ATP synthase inhibitor). The results showed that after the addition of exogenous Oligomycin A, the color of the fruiting body cap became lighter, further indicating that ATP plays an important role in the formation of cap color (Figure 9).

## 4. Discussion

H_2_O_2_ is a key signaling molecule in cell proliferation, development, tissue differentiation, and environmental responses in various organisms. Under stress, H_2_O_2_, as the main component of ROS, can cause cell damage. For example, under heat stress, the H_2_O_2_ content of *P. ostreatus* mycelia significantly increases, causing damage to the cell wall and significantly inhibiting the growth rate of the mycelia [33,39]. On the other hand, as a signaling molecule, H_2_O_2_ also plays an important role in the growth and development of mushrooms. For example, in *G. lucidum*, H_2_O_2_ can regulate the synthesis of the active substance triterpenoid compounds [30]. In *F. filiformis*, ROS (O_2_^−^/H_2_O_2_) redistribution mediated by NADPH oxidase and MnSODs is linked to the gradient elongation of the stipe [40]. Our team’s previous research found that H_2_O_2_ can be significantly influenced by NO regulation to significantly affect the formation rate of *P. ostreatus* primordia [32]. Currently, increasing evidence suggests that H_2_O_2_ can act as a signaling molecule to upregulate the expression of many genes activated under environmental stress [41]. In plants, H_2_O_2_, as a central redox-signaling molecule in physiological oxidative stress, can induce the expression of 113 genes during oxidative stress, which are involved in cellular rescue and defense processes [42]. The growth and development of edible mushrooms is a very complex process and can also be subjected to various environmental stresses. For example, *P. ostreatus* is subjected to temperature and light stress during the mushroom production process. Our previous research found that H_2_O_2_ is distributed differently in different parts of the *P. ostreatus* fruiting body and significantly accumulates in the cap [37]. In this study, the results further found that when H_2_O_2_ in the cap is removed by its scavenger (DMTU), the cap color turns white, indicating the key role of H_2_O_2_ in the formation of cap color. Previous studies have shown that the biological function of the browning of *F. filiformis* could act as a protective mechanism response to the damage of cell integrity, and the enzymatic dopa melanin pathway could contribute to the browning mechanism [43]. In addition, the browning of *F. filiformis* may be caused by DNA damage, proteolysis, and oxidative photosynthesis process, which together lead to the damage of cell integrity and then trigger the biosynthesis of melanin [44]. Therefore, it is speculated that the melanin in the cap of *P. ostreatus* may also be a protective mechanism against environmental stress. Further analysis by RNA-seq showed that a link between oxidative phosphorylation and the formation of the *P. ostreatus* cap color existed. Our results are similar to those of previous studies, suggesting that oxidative phosphorylation may be involved in the synthesis of melanin.

The application of omics technology in the study of edible mushrooms provides a scientific basis for exploring regulatory mechanisms. In animals and plants, research on oxidative phosphorylation and color is also in preliminary progress. The formation of red color is an obvious sign of the beginning of strawberry fruit ripening. Transcriptome analysis showed that a series of metabolic changes occurred in the green-white-red phase. DEGs change from variable to less in this process, and oxidative phosphorylation plays an important role in regulating the ripening process [45]. Previous studies have shown that oxidative phosphorylation has a significant effect on the color of black-cut beef [46]. Moreover, efficient energy production and metabolism pathways (i.e., oxidative phosphorylation and the citric acid cycle) may promote the expression of key genes and proteins in the carotenoid selective transport system of the silkworm [47]. In *Crassostrea gigas* with orange and black shells, RNA high-throughput sequencing data showed that DEGs could be significantly enriched in “oxidative phosphorylation” related to ATP synthesis, suggesting that they might participate in pigment deposition [48]. In this study, 614 DEGs specifically regulated by H_2_O_2_ were preliminarily screened by RNA-seq, and only the oxidative phosphorylation pathway was significantly enriched by KEGG-enrichment analysis. This study preliminarily clarified the possible regulatory pathway of H_2_O_2_ as a signaling molecule in *P. ostreatus* cap color formation.

Oxidative phosphorylation plays a key role in the growth and development of organisms: it is the fundamental source of carbon skeletons and the driving force of biochemical reactions. In this study, further analysis revealed that 23 DEGs in the oxidative phosphate pathway were enriched in Complexes I, III, IV, and V of the respiratory chain. Therefore, H_2_O_2_ may affect the formation of fungal cap color by regulating the respiratory chain. The addition of exogenous inhibitors is a commonly used experimental method in many studies. In this study, it was found that the addition of respiratory chain uncoupling agents resulted in a lighter cap color. Further evidence suggests that ATP synthesis may play a crucial role in the formation of cap color in *P. ostreatus*. Moreover, the addition of exogenous ATP synthase inhibitors further demonstrated the importance of ATP synthesis. Previous studies have shown that the postharvest browning of *A. bisporus* is closely related to energy metabolism [49]. It is speculated that ATP may play an important role in color formation.

H_2_O_2_, as a signaling molecule, may have complex and multiple signal transduction mechanisms in regulating the color synthesis process of the *P. ostreatus* cap. At present, our study showed that H_2_O_2_ may regulate the formation of cap color by promoting ATP synthesis. In the future, it is necessary to further explore the molecular mechanism involved in the downstream stages of the H_2_O_2_ pathway and provide a better understanding of the color formation mechanism of mushrooms. Superoxide dismutase and catalase are important enzymes regulating H_2_O_2_ content. Respiratory chain Complexes I and III are the main sites of H_2_O_2_ production. Therefore, in future work, it will be necessary to explore the functions of these key genes and further clarify the molecular mechanism by which H_2_O_2_ regulates cap color formation.

## 5. Conclusions

In conclusion, the color of the *P. ostreatus* cap is an important agricultural trait, and its formation process is regulated by multiple genes with complex regulatory mechanisms. H_2_O_2_ plays an important regulatory role in *P. ostreatus* cap color formation. In this study, RNA-seq was used to comprehensively understand the molecular basis and regulatory mechanism by which H_2_O_2_ regulates cap color formation. The study showed that H_2_O_2_ may affect cap color formation by regulating the metabolic pathway of oxidative phosphorylation. The 23 DEGs encoding Complexes I, III, IV, and V subunits in the respiratory chain can be significantly regulated by H_2_O_2_. Furthermore, ATP synthesis was found to be the key factor affecting the formation of cap color.

## Figures and Tables

**Figure 1 jof-09-00823-f001:**
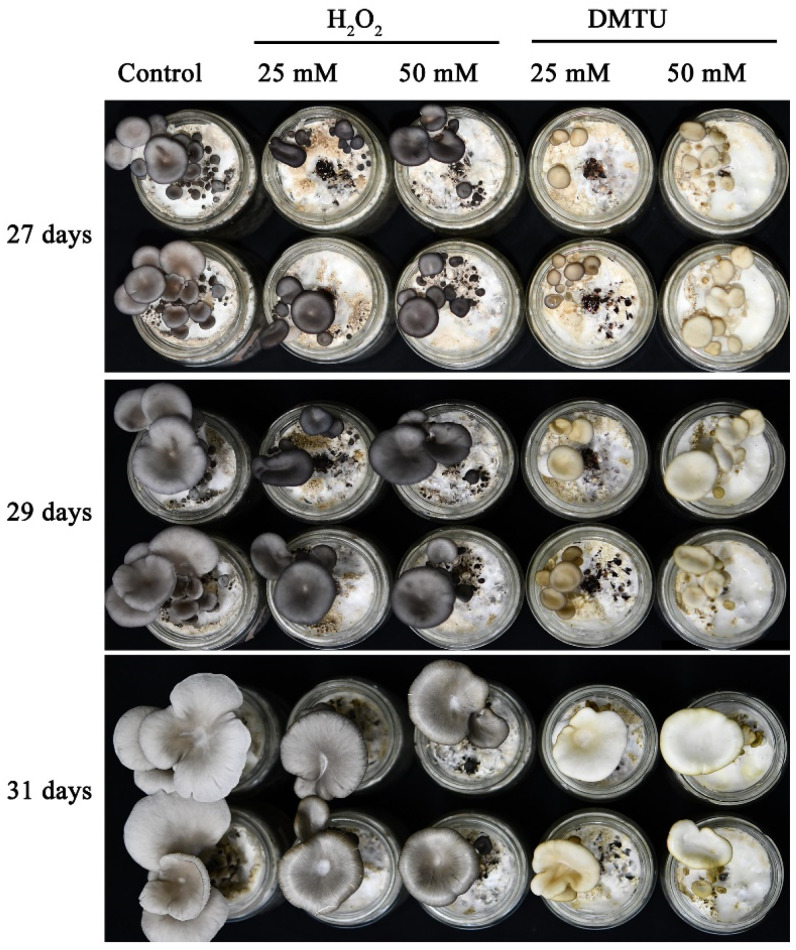
H_2_O_2_ regulates cap color formation in *P. ostreatus*.

**Figure 2 jof-09-00823-f002:**
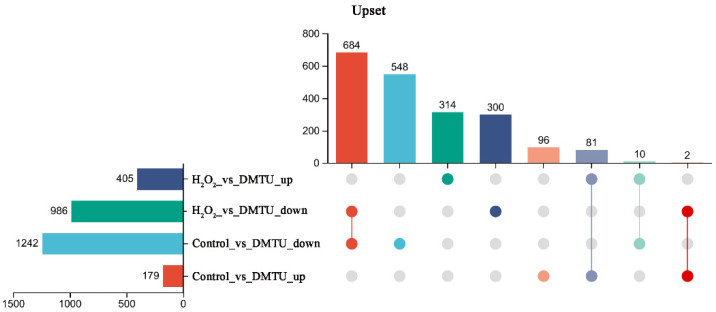
Number of upregulated and downregulated DEGs among the control, H_2_O_2_, and DMTU groups.

**Figure 3 jof-09-00823-f003:**
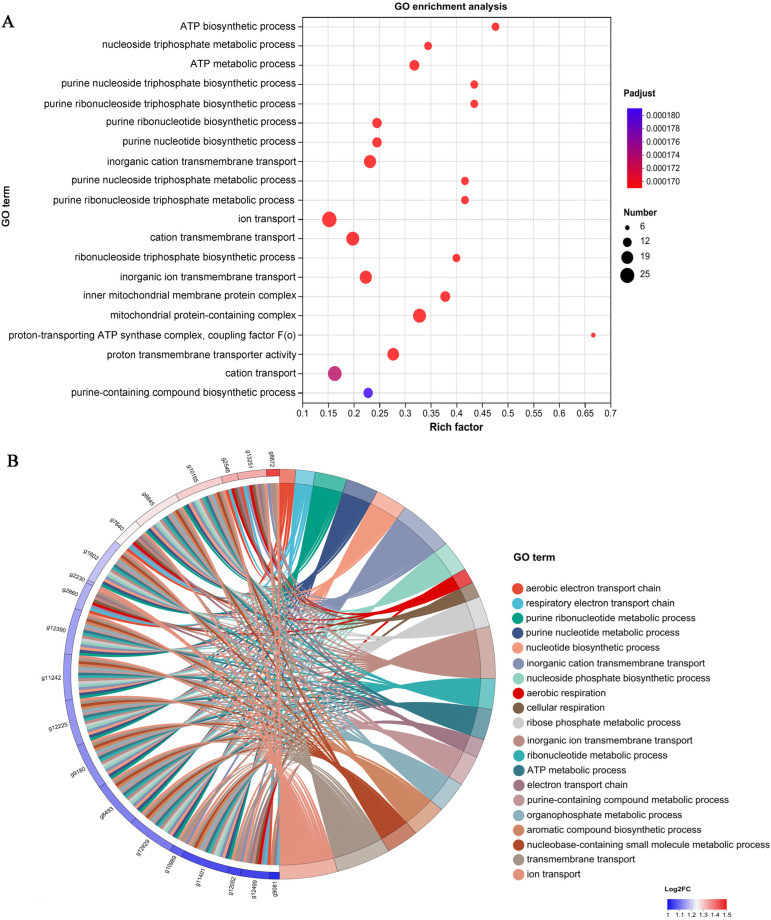
Significantly enriched pathways and GO enrichment chord chart for the DEGs. (**A**) GO enrichment analysis of DEGs from the H_2_O_2_ vs. DMTU comparison. (**B**) GO enrichment chord chart of DEGs from the H_2_O_2_ vs. DMTU comparison.

**Figure 4 jof-09-00823-f004:**
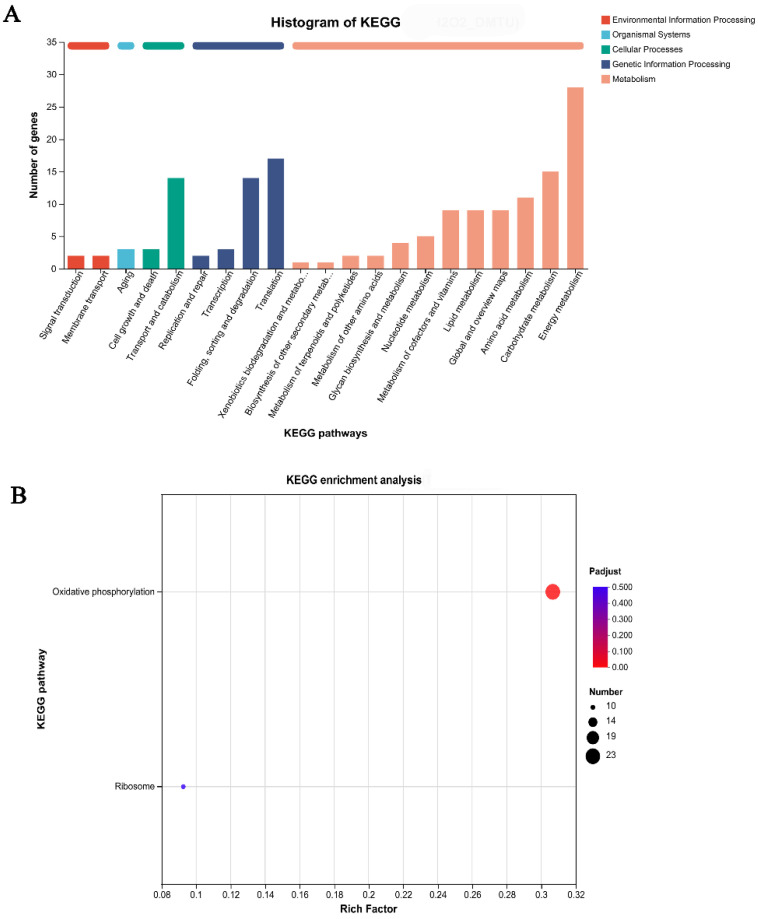
Enriched KEGG pathways of identified DEGs. (**A**) Enrichment of function; (**B**) Enrichment of metabolic pathways.

**Figure 5 jof-09-00823-f005:**
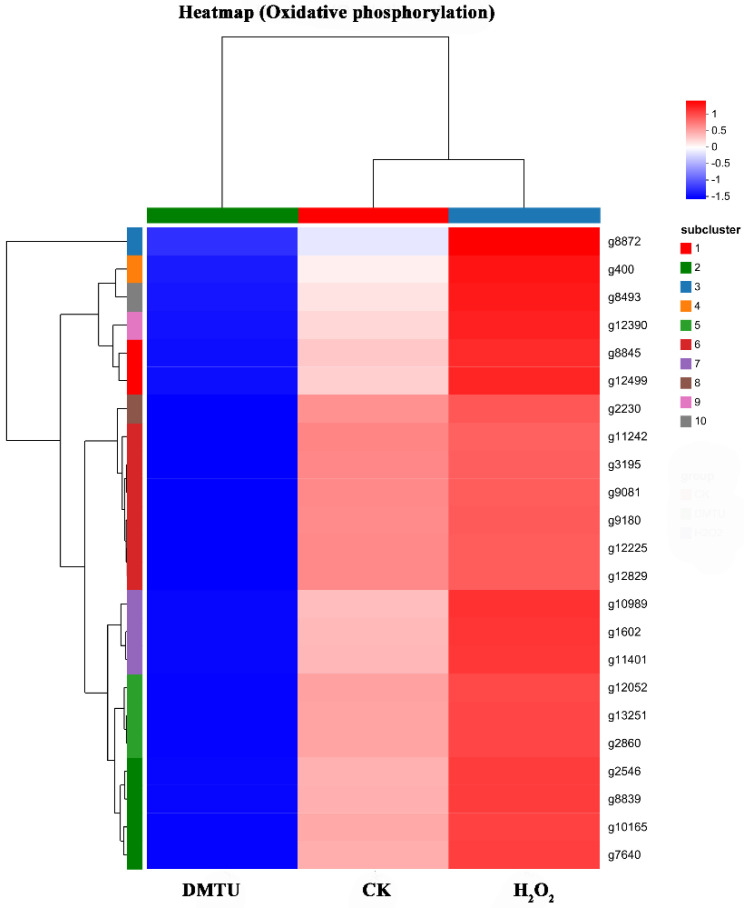
Heatmap of oxidative phosphorylation consisting of the 23 DEGs.

**Figure 6 jof-09-00823-f006:**
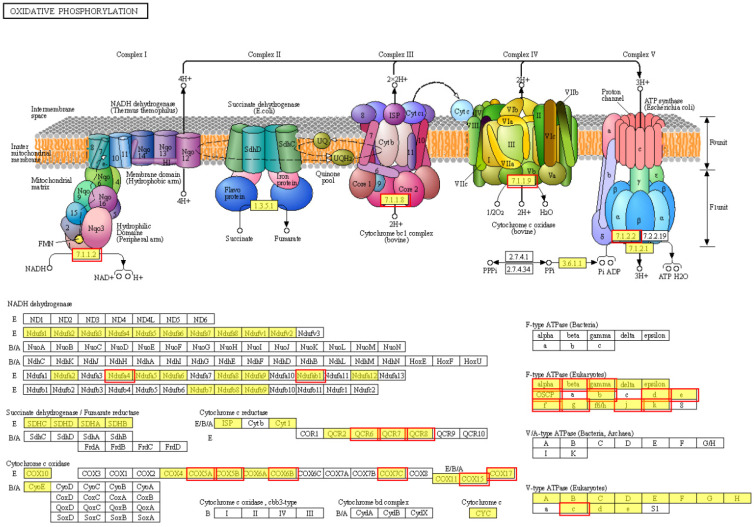
Regulatory effect of H_2_O_2_ on the oxidative phosphorylation metabolic pathway. The red rectangle represents the 23 DEGs of interest and their encoded subunits in the oxidative phosphorylation metabolic pathway.

**Figure 7 jof-09-00823-f007:**
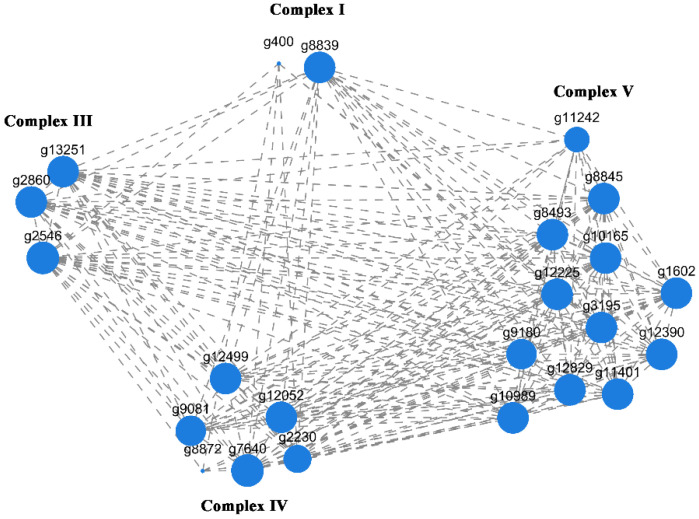
Correlation between the expression of 23 DEGs. Each node in the graph represents a gene, and the connections between nodes represent a correlation in gene expression. The larger the node, the number of genes with expression correlation with other genes.

**Figure 8 jof-09-00823-f008:**
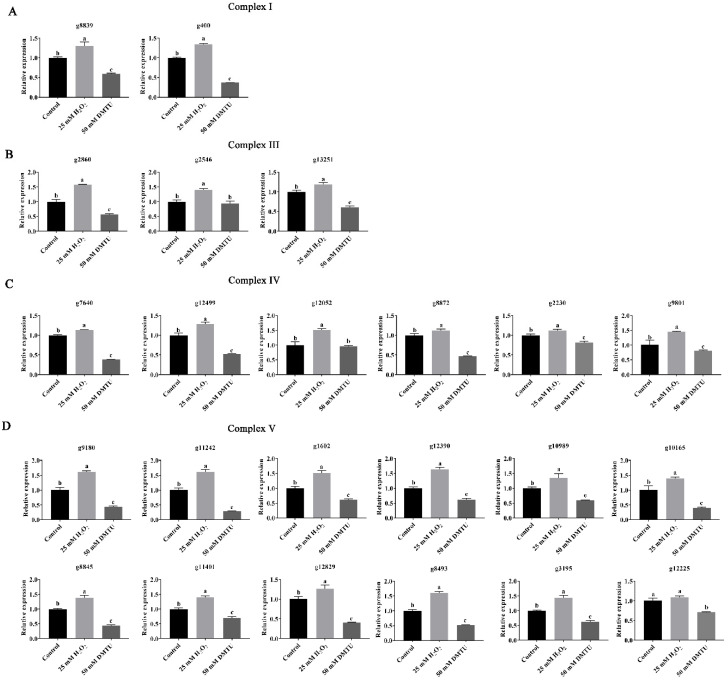
H_2_O_2_ regulates the expression of complex subunit-encoding genes in the respiratory chain. (**A**) Validation of DEGs by qPCR in Complex I; (**B**) Validation of DEGs by qPCR Complex III; (**C**) Validation of DEGs by qPCR Complex IV. (**D**) Validation of DEGs by qPCR Complex V. Different letters indicate significant differences for the comparison of samples (*p* < 0.05 according to Duncan’s test).

**Figure 9 jof-09-00823-f009:**
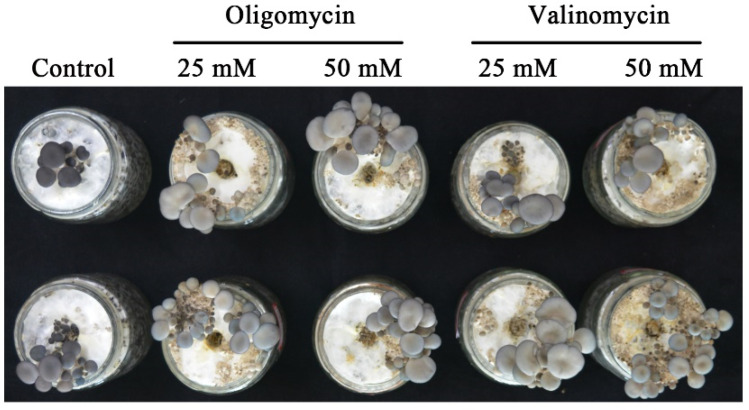
ATP synthesis affects the formation of cap color.

**Table 1 jof-09-00823-t001:** The functions of 23 DEGs in oxidative phosphorylation.

Gene ID	Gene Function
*g10165*	ATP4 subunit B of the stator stalk of mitochondrial F1F0 ATP synthase
*g2546*	Cytochrome bc1 complex subunit 7
*g7640*	Cytochrome c oxidase subunit 6
*g8839*	NdufA4 NADH dehydrogenase 1 ɑ subcomplex
*g10989*	V-type proton ATPase 16 kDa proteolipid subunit 2
*g11401*	ATP17 subunit F of the F0 sector of mitochondrial F1F0 ATP synthase
*g1602*	ATP3 γ subunit of the F1 sector of mitochondrial F1F0 ATP synthase
*g11242*	ATP synthase F1 β subunit
*g12225*	ATP synthase E chain-domain-containing protein
*g12829*	ATP20 subunit g of the mitochondrial F1F0 ATP synthase
*g3195*	Hypothetical protein
*g9081*	Cytochrome c oxidase assembly protein cox15
*g9180*	ATP synthase F1 ɑ subunit
*g12052*	Cytochrome c oxidase subunit vib
*g13251*	Ubiquinol-cytochrome c reductase complex subunit 8
*g2860*	Ubiquinol-cytochrome-c reductase subunit 6
*g12390*	ATP synthase subunit 5
*g12499*	Cytochrome c oxidase subunit 4
*g8845*	ATP synthase d subunit
*g2230*	Cytochrome c oxidase copper chaperone
*g400*	Acyl carrier protein
*g8493*	ATP18 subunit j of the mitochondrial F1F0 ATP synthase
*g8872*	Hypothetical protein

## Data Availability

The data of all results in this study are included in the manuscript and Appendix A. If necessary, the data can be obtained by contacting the corresponding author.

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
