# Peer review of "Transcriptome Analysis Revealed That Hydrogen Peroxide-Regulated Oxidative Phosphorylation Plays an Important Role in the Formation of Pleurotus ostreatus Cap Color"

_jof, 2023, doi:10.3390/jof9080823_

Round 1

Reviewer 1 Report

In this paper the author studied the effect of external H2O2 application on expression of P. ostreatus during fruiting bodies formation with emphasis on cap dark color.  Global transcriptomic changes in response to H2O2 or DMTU treatments were analyzed and described. It was concluded that  ATP synthesis and metabolism may be related to the formation of cap color. H2O2 can affect the formation of cap color by upregulating the expression of key genes in the   respiratory chain. H2O2 participates in the regulation of cap color   by regulating the expression of genes encoding complex I, III, IV, and V subunits in the   respiratory chain. The conclusions are mostly based on the omics data. However, the role of ATP was verified by applying exogenous valinomycin or oligomycin, resulting in lighter color.

The paper is interesting and provides new transcriptomic data related to the mushroom biology. The bioinformatics analyses were well performed and shown in the figures. I think that the authors could discuss how to affect the concentration of the internal H2O2 in the cap to achieve similar result to the external application. What are the other effects of H2O2, beyond signaling, as it is a substrate to variety of Pleurotus enzymes.  Form the applied side, breeding of dark-colored strains, what genes can be manipulated in relation to internal H2O2 synthesis. I also assume that ATP synthesis and metabolism can affect other physiological and developmental processes – is it feasible to manipulate this pathway and achieve specific effect.  

  Specific point

 line  346 –"bacterial cap" – should it be  fungal?

Reviewer 2 Report

Review on jof-2478026

The manuscript “Transcriptome analysis revealed that hydrogen peroxide-regulated oxidative phosphorylation plays an important role in the formation of Pleurotus ostreatus cap color” presents the in-depth analysis of the transcriptome of H2O2- and DMTU-treated fruiting bodies to elucidate the molecular basis of H2O2-induced cap darkening. The manuscript is well written and interesting to read. The experiments were conducted carefully and the results should be of interest for the readers of the “Journal of Fungi”.

The major problem is the conclusion (first time in l. 232) that since genes connected to oxidative phosphorylation and energy metabolism are upregulated after treatment with H2O2, this has to be the reason for the darkening of cap colour. After all, a high number of different genes was upregulated after treatment with H2O2, any of which might be responsible for the change in colour. In my opinion, the experiments using valinomycin and oligomycin A are the only proof that ATP synthesis is more or less directly involved in cap colouring, but even here, further ATP-dependent processes might be involved that remain unnamed. Please carefully check your conclusions and rewrite the relevant sections of the results, discussion and conclusion.

Other than this, some minor points have to be amended:

L. 17: DEGs are not a commonly used abbreviation. I would prefer the full name in the abstract as some people do not read the whole manuscript.

Introduction:

-        I am missing information on why the cap colour is of interest at all.

-        Ll. 50-70 reads like a list of information instead of a text. The aim of this paragraph remains unclear.

-        L. 80 repeats the information from l. 68. It is then again repeated in l. 99.

-        Where is the reference for the previous research mentioned in ll. 98-99?

M&M:

-        L. 121: CK is control?

Results:

-        3.1: What about the results gained with DMTU? What do they indicate? The manuscript also fails to mention/discuss what exactly DMTU does at all (apart from brightening the cap colour).

-        Ll. 180-186: What about H2O2 vs. the control? Why is this not shown at all?

-        Ll. 186-189: How were the 614 genes chosen for further investigation? After all, there were only 405 genes upregulated after addition of hydrogen peroxide in contrast to DMTU.

-        Fig. 2: The legend describes the left part of the figure. No description is given for the right part of the figure and it is also not mentioned in the text as far as I understand. What is shown here?

-        Ll. 196-198: The choice of pathways is not comprehensible from Fig. 3a. Why are ATP metabolic processes mentioned, but not nucleoside triphosphate metabolic processes?

-        Fig 3: Both parts are extremely small. Even with zooming in, I cannot read anything on the legend of 3b at all.

-        Ll. 223-225: The two sentences seem very repetitive.

-        Fig 6: I do not see the necessity for this figure. What do you want to show? It certainly does not show the ‘regulatory effect of H2O2’ even though the legend claims this. Or is the legend just incomplete and the red rectangles are the 23 genes of interest?

-        Fig. 7: Why are there no points for g400 (I) and g8872 (IV)? And what exactly was measured here for the correlation?

-        L. 266: Should it be g2546 instead of 2549?

-        Fig. 8: As you did the significance testing according to the text: Please add the respective asterisks etc. to the graph for an easy overview.

-        Ll. 288-289: The sentence is incomplete.

Discussion:

-        Ll. 303-309: Is the information regarding plant metabolism really helpful here?

-        L. 320: What does (received) mean?

The manuscript is well written apart from minor problems that are mentioned above.

Round 2

Reviewer 2 Report

The revision process was done very thoroughly and well. I think the manuscript is now ready to be published after three minor adjustments:

- Please make the legends of figure 6 and 7 detailed enough to be understandable. The information about the red rectangles in figure 6 and the size of the knots in figure 7 that you gave me in your answer should be in the respective figure legends.

 to - The new insert in the introduction, ll. 67-71, needs a reference.

Everything else is good to be published.

The english language is fine.
